# High Gain, Low Noise and Power Transimpedance Amplifier Based on Second Generation Voltage Conveyor in 65 nm CMOS Technology

**DOI:** 10.3390/s22165997

**Published:** 2022-08-11

**Authors:** José C. García-Montesdeoca, Juan A. Montiel-Nelson, Javier Sosa

**Affiliations:** Institute for Applied Microelectronics, University of Las Palmas de Gran Canaria, 35017 Las Palmas de Gran Canaria, Spain

**Keywords:** CMOS technology, down converter, voltage conveyor, low supply voltage, output low swing, low energy consumption, high bandwidth, signal processing

## Abstract

A transimpedance amplifier (TIA) based on a voltage conveyor structure designed for high gain, low noise, low distortion, and low power consumption is presented in this work. Following a second-generation voltage conveyor topology, the current and voltage blocks are a regulated cascode amplifier and a down converter buffer, respectively. The proposed voltage buffer is designed for low distortion and low power consumption, whereas the regulated cascode is designed for low noise and high gain. The resulting TIA was fabricated in 65 nm CMOS technology for logic and mixed-mode designs, using low-threshold voltage transistors and a supply voltage of ±1.2 V. It exhibited a 52 dBΩ transimpedance gain and a 1.1 GHz bandwidth, consuming 55.3 mW using a ±1.2 V supply. Our preamplifier stage, based on a regulated cascode, was designed considering detector capacitance, bonding wire, and packaging capacitance. The voltage buffer was designed for low-power consumption and low distortion. The measured input-referred noise of the TIA was 22 pA/√Hz. The obtained total harmonic distortion of the TIA was close to 5%. In addition, the group delay is constant for the considered bandwidth. Comparisons against published results in terms of area (A), power consumption (P), bandwidth (BW), transimpedance gain (G), and noise (N) are were performed. Both figures of merit FoMs—the ratio √ (G × BW) and P × A—and FoM/N values demostrated the advantages of the proposed approach.

## 1. Introduction

A dramatic demand for charge detection in commercial and scientific applications, such as mass spectrometry, DNA analysis, imaging, and nuclear science, among others, has been reported in the published literature [1,2]. For fast detection and portable systems, the goal is to obtain low power consumption and reduced-area systems, mainly through the use of application-specific integrated circuit (ASIC) technology. Charge-sensitive amplifiers and shaping amplifiers [3,4] have been used to design front-end receiver circuits and systems based on ASIC implementations. In the sensor related literature, some works have focused on noise reduction techniques and improving performance in terms of resolution [5,6,7] in the readout ASICs. In addition, further improvements, such as boosting the signal acquisition speed, reducing temporal delays associated with the charge integration time in the charge-sensitive amplifier (CSA), and the filtering of the output voltage signal in the shaping time of the CSA, have been published.

Charge amplifier applications require high open-loop gain (>100 dB) but CMOS circuits present low intrinsic gain in spite of their low fabrication cost and their amenability for large-scale integration. A transimpedance amplifier usually confers limited bandwidth, noise and sensitivity values on the whole system [8,9], particularly when it is employed at the first stage of the analog processing chain, i.e., as the input of the receiver’s front ends to directly convert the charge generated by photodiodes or photomultipliers into voltage signals. This is associated with the large input of the parasitic capacitanceof the photodiode (PD), which is connected as an input signal to the charge amplifier based on TIA, and represents the dominant pole of the system. Thus, a low input resistance increases the bandwidth of the PD-TIA system. Furthermore, the system’s sensitivity at high speed is reducedif the parasitic capacitance at the input is high. Additionally, the performance of the complete system can deteriorate due to the input-referred noise of the TIA. The use oflarger devices helps to reduce the noise but this increases the parasitic capacitance and power consumption. Therefore, it is necessary to satisfy the constraints for the design of a low-power low-noise TIA operating at a high speed and providing a large transimpedance gain. To date, the sensor has been connected through the analog PADS of the die, comprising a mixed-signal circuit. In the near future, emergent technologies for integrated systems will include photodiodes as sensor blocks.

Two topologies of amplifiers are used in front-end charge detectors: schemes based on a common gate (base) configuration, and architectures employing high-gain amplifiers with shunt-shunt feedback. These structures offer different noise performance and it is imperative to select the best circuit for a given application and with the consideration of low input impedance as well. Amplifiers based on the common gate topology are chosen for TIA systems because they have a low input impedance and provide the best solution in contrast with a high-gain amplifier enclosed with shunt-shunt feedback, which presents an input impedance that is dependent on process variations [2].

Voltage conveyors are flexible building blocks for active circuit synthesis. A first-generation voltage conveyors (VCIs) include a CCCS (Current-Controlled Current Source) and two VCVSs (Voltage-Controlled Voltage Sources) [10], whereas second-generation voltage conveyors (VCII) are formed by a CCCS and a VCVS [11].The third generation voltage conveyor (VCIII) [12] is a special case of a three-port immittance convertor, consisting of summing-voltage immittance convertors (SVICs), which are obtained using resistors and operational amplifiers. A VCII is also a helpful circuit for the processing of electrical signals coming from silicon photomultipliers compared to a traditional voltage operational amplifier [13].

The operation of a voltage conveyor relies on conveying voltage signals and it is designed with a current buffer, followed by a voltage buffer. In fact, the current buffer acts as a TIA and the connected voltage buffer provides a further impedance-matching function. For example, in [14], the class-AB technique was used in both input and output stages to obtain a high-performance VCII.

Second-order filters are known as biquadratic filters. In [15], corresponding filter voltage transfer functions were obtained using appropriate admittance choices. These biquads are electronically tunable and the generalized topology is based on two voltage conveyors and six admittances. Circuits using VCII show high frequency performance and circuits operating in class-AB have low quiescent current and high current drive capability. In [16] a voltage integrator which meets the requirements of low-power consumption and high drive capability for a specified transient performance was presented. This VCII solution uses a topology with two class-AB flipped voltage follower (FVF) circuits. Finally, the authors in [13] proposed an electronically tunable grounded capacitor multiplier, which was able to emulate large value capacitors and to reduce the chip area. In addition, this capacitor multiplier relied on VCII and operated in the subthreshold region with low voltage and low power consumption. A VCII working as a signal conditioner amplifier (SCVC) for a photomultiplier sensor was introduced. The main advantage of this recent approach is that the proposed circuit is implemented using commercial off-the-shelf (COTS) devices, achieving a bandwidth of 106 MHz, transimpedance gain of 42 dBΩ, and a power consumption of 200 mW. Recently, it became possible to use fully depleted silicon on insulator (FD-SOI) CMOS technology to optimize the performance of voltage conveyors and to decrease the complexity of the circuit and power consumption. In [17], the authors introduced a second-order bandpass filter using 28 nm FD-SOI technology.

In this paper, we introduce a transimpedance amplifier based on a second-generation voltage conveyor topology, which can be used in high-performance signal processing systems for silicon photomultiplier sensor interfacing. The key contributions of this paper are:A current buffer (CB) based on a regulated cascode topology is optimized for both noise and bandwidth gain. The input-referred noise is analyzed when a detector capacitance $C_{d}$ is used at the input of the CB and both the bonding wire $L_{b}$ and packaging capacitances are considered. Simulation and measurement results show a noise capacitive peaking which is diminished by the input inductance $L_{b}$.A voltage buffer (VB) is designed for low distortion and low power consumption, as well as an impedance adapter. The CB-VB set acts as a VCII, and its low distortion characteristics demonstrate its usefulness as a linear voltage amplifier.The TIA was implemented in CMOS using a 65 nm logic and mixed-mode technology process. From twelve fabricated circuits, the measurement results were 52 dB transimpedance gain and 1.1 GHz of bandwidth, together with 22 pA/√Hz of input-referred noise and 55.3 mW of power consumption.Comparisons against published results demonstrated improvements in terms of noise, gain-bandwidth, and the energy-delay product.

The rest of this paper is organized as follows. Section 2 introduces the transimpedance amplifier design based on the VCII topology. The preamplifier stage is a current buffer which has been designed to improve gain and noise performance. Noise is analyzed in detail and measurements based on an ASIC implemented for charge detection are discussed. In addition, a voltage buffer is proposed to obtain low power consumption and low signal distortion. Section 3 gives details of the layout and chip implementation. Section 4 provides measurements for a population of twelve fabricated dies. In addition, the results of comparisons with the published literature are presented. Finally, in Section 5, conclusions are drawn.

## 2. TIA Based on VCII

Figure 1 illustrates the use of a VCII block as a transimpedance amplifier in the front end of a receiver to directly convert charges generated by photomultipliers or photodiodes into a voltage signal. The VCII, in this work, is placed at the front-end of an ASIC which is implemented in a CMOS process of 65 nm for a logic and mixed-mode design with transistors of low-leakage currents and that use copper metallization with 7 metal levels and low-K dielectrics. This 65 nm CMOS process provides enables the design of low-threshold voltage transistors and IO transistors of 5.0 nm gate oxide for a supplied voltage of 2.5 V.

The integrated IP ASIC is suitable for use both in sensor interfacing and in post-processing at a digital level as an SOC. The input stage of the ASIC (CB in Figure 1) acts as a preamplifier, whereas VB is the voltage buffer. The CB topology is based on a regulated cascode (super common gate), which is optimized to improve the input-referred noise, while preserving the bandwidth. Figure 2 shows a schematic of a regulated cascode amplifier driven by a differential amplifier and at the output $v_{o}\left(s\right)$ is a transfer function and $v_{t}\left(s\right)$ is a voltage buffer or filter. A regulated cascode (see Figure 2) is widely used, especially in particle physics designs, as the input impedance is small, and lower than the impedance of the common gate amplifier by a factor close to the gain of the booster amplifier—this is depicted as stages MN2-MP2 in Figure 2.

Figure 3 shows a circuit schematic of the implementation of the VCII presented in [13] (SCVC). The original approach uses commercial devices, and the circuital structures used to implement the current sources are not directly implementable on integrated circuit CMOS technologies. For our purposes, we have modified this design to replace those current sources with diode-connected transistors. Through this approach, the use of a VCII system in current CMOS technologies can be obtained.

In this paper, we use this approach as a reference topology. It consists of a current buffer, where the input current incoming from node Y is mirrored on node X. Additionally, the second stage is a voltage buffer, implementing voltage mirroring between nodes X and Z. In this way, the VCII structure operates as a transimpedance amplifier between nodes Y and Z with low input and output impedances. Note that the transistor MN8 uses a technical method called a flipped voltage follower (FVF), reported in [16], to obtain a fixed drain-source voltage. In [13], capacitances C1 and C2 are 1 μF AC coupling capacitors for the input node X and output node Z, respectively. Regarding resistors R1 and R2, we assume a value of 100 kΩ, whereas the Rx and Rz loading resistors are equal to 10 kΩ and 10 MΩ, respectively.

### 2.1. Preamplifier Design and Noise Analysis

Following the small signal model of the regulated cascode circuit—including noise sources; see Figure 4—with the source of MN4 as an input and MN2 placed in the feedback or booster amplifier, the input impedance is $1/(gm_{N4}G_{fb})$, where $G_{fb}$ is the feedback gain, i.e., $\approx gm_{N2}req_{2}$, where $req_{2}$ is the equivalent resistance at node 2. This topology partially solves the tradeoff between noise performance and input impedance in comparison with the common gate topology. In Figure 4, the transimpedance $v_{o}\left(s\right)/i_{in}\left(s\right)$ is calculated, assuming that the noise current generators of MN4 and MN2 are open. Assuming that $Z_{i}\left(s\right)\approx C_{d}$, $rds_{N4}>>R_{L}$ and $gm_{N2}(rds_{N2}||rin_{P2})>>1$, then
(1)$$\frac{v_{o}\left(s\right)}{i_{in}\left(s\right)}\approx \frac{Z_{L}}{\left[1+s\tau_{1}\left(1+a\right)+s^{2}\tau_{1}\tau_{2}\right]},$$
where $\tau_{1}=C_{d}/\left(gm_{N4}gm_{N2}req_{2}\right)$, $\tau_{2}=req_{2}C_{12}$, $a=gm_{N2}req_{2}C_{12}/C_{d}$, $req_{2}=rds_{N2}\left|\right|rin_{P2}$, and $C_{12}=Cgd_{N2}+Cgs_{N4}$. In Equation (Equation 1), $gm_{N4}\;\mathrm{and}\;gm_{N2}$ are channel transconductance of transistors MN4 and MN2, respectively; $rds_{N2}$ is the drain-to-source resistance of transistor MN2, and $rin_{P2}$ is the input impedance of transistor MP2. As shown, the input capacitance of the detector, $C_{d}$, reduces the bandwidth of the regulated cascode. The gain is modulated by transconductances of the MN2 and MN4 transistors.

In analyzing the input-referred noise of the TIA preamplifier, the input capacitance, i.e., the detector capacitance $C_{d}$, increases this noise, as demonstrated in the following. In addition, when the photodetector is connected to an integrated TIA, the detector capacitance $C_{d}$, the inductance of the bonding wires $L_{b}$, and the capacitance of the packaging and die pads $C_{p}$ contributes to the input-referred noise of the TIA. In fact, in exploiting the inductive peaking technique, the TIA successfully reduces the input = referred noise and improves the system gain and bandwidth, as was presented in [18]. In Figure 2, the impedance parasites at the input of the amplifier, i.e., the capacitance $C_{d}$ of the detector (up to 5 pF for a photomultiplier tube and 35–300 pF for a silicon photomultiplier [19]); the inductance of the bonding wire ($L_{b}$) for an encapsulated TIA; ≈1 nH for a curved wire of 1.5 mm in length; and the pad capacitance ($C_{p}$) for the fabricated die (≈0.5 pF) were included.

In Figure 4 we present a simplified small-signal model of the preamplifier, including the noise currents due to the transistors MN2 and MN4. MN2 contributes as a source of series noise and MN4 as a source of parallel noise. Noise optimization is carried out by achieving the proper balancing of the series and parallel noise contributions from transistors MN2 and MN4, respectively. In Appendix A, Kirchhoff’s voltage equations for nodes 0, 1, 2 and 3 of the simplified small-signal model are presented.

For the small-signal circuit, the transimpedance transfer function is expressed as:(2)$$\frac{v_{o}\left(s\right)}{i_{in}\left(s\right)}=\frac{v_{o}\left(s\right)}{v_{1}\left(s\right)}\;\frac{v_{1}\left(s\right)}{i_{i}\left(s\right)}\;\frac{i_{i}\left(s\right)}{i_{in}\left(s\right)},$$
where $v_{o}\left(s\right)/i_{in}\left(s\right)$ is the equivalent input impedance $Z_{i}\left(s\right)$, and $i_{i}\left(s\right)/i_{in}\left(s\right)$ is the current transfer function:(3)$$\frac{i_{i}\left(s\right)}{i_{in}\left(s\right)}=\frac{1}{1+sZ_{i}\left(s\right)\left(C_{p}+C_{d}\right)+s^{2}L_{b}C_{d}+s^{3}Z_{i}\left(s\right)C_{p}C_{d}L_{b}}.$$

For a regulated cascode:(4)$$Z_{i}\left(s\right)\approx \frac{1}{g_{mN2}g_{mN4}r_{inP2}}.$$

In comparison of the input impedance of the MN4 transistor, $1/g_{mN4}$, the input impedance of the regulated cascode is lower by a factor $1/(g_{mN2}r_{inN2})$—the gain of the booster amplifier.

Therefore,
(5)$$\frac{v_{o}\left(s\right)}{i_{in}\left(s\right)}\approx \frac{v_{o}\left(s\right)}{v_{1}\left(s\right)}\;\frac{Z_{i}\left(s\right)}{1+s^{2}L_{b}C_{d}},$$
because $sZ_{i}\left(s\right)\left(C_{p}+C_{d}\right)+s^{3}Z_{i}\left(s\right)C_{p}C_{d}L_{b}$ are neglected with respect to $1+s^{2}L_{b}C_{d}$. Thus, the transimpedance gain $v_{o}\left(s\right)/i_{in}\left(s\right)$ of the amplifier is modified by the transfer function $1/(1+s^{2}L_{b}C_{d})$.

To calculate the equivalent input noise contributed by MN4, the noise generator $i_{n4}$ is turned on, whereas the rest of the current sources are turned off. Consequently, by solving (Equation 5), $v_{o}\left(s\right)$ in terms of $i_{in}\left(s\right)$,
(6)$$\frac{v_{o}\left(s\right)}{i_{in}\left(s\right)}\approx \frac{r_{ds4}R_{L}}{r_{ds4}+r_{ds5}}\left(\frac{1+sC_{d}r_{ds5}}{1+sr_{ds5}C_{d}\frac{r_{ds4}}{r_{ds4}+r_{ds5}}}\right),$$
when only the capacitance $C_{d}$ is considered and the bonding wire inductance is neglected. Note that $C_{d}$ is greater than 5 pF when the bonding wire inductance $L_{b}$ is 1 nH, approximately, for a curved wire of 1.5 mm in length. By increasing the input inductance $L_{b}$, some noise suppression is achieved. The input noise transfer function introduces one zero and one pole at frequencies of $1/(r_{ds5}C_{d})$ and $\left(r_{ds4}+r_{ds5}\right)/\left(r_{ds4}r_{ds5}C_{d}\right)$, respectively. Therefore, the contribution of $C_{d}$ at the input stage of impedanceincreases the equivalent noise contribution by MN4. However, when $L_{b}$ is considered, the relationship $v_{o}\left(s\right)/i_{in4}\left(s\right)$ is given by
(7)$$\frac{v_{o}\left(s\right)}{i_{in}\left(s\right)}\approx \frac{r_{ds4}R_{L}}{r_{ds4}+r_{ds5}}\left(\frac{s^{2}C_{d}L_{b}+sC_{d}r_{ds5}+1}{s^{2}C_{d}L_{b}+C_{d}\frac{r_{ds4}r_{ds5}}{r_{ds4}+r_{ds5}}+1}\right).$$

Transfer function (Equation 7) contains two zeroes and two poles at the same frequency of $1/(C_{d}L_{b})$ (rad/s). Hence, bonding wire inductance reduces the equivalent noise from MN4.

The TIA preamplifier (see the schematic in Figure 2) has been designed not only for gain × bandwidth performance but also considering the input-referred noise. The effect of the detector capacitance $C_{d}$ is considered, as well as the bond wire inductance $L_{b}$ and packaging capacitance $C_{p}$. Figure 5 introduces the simulated and measured input-referred noise of the complete transimpedance amplifier (DIE) and the effects of $C_{d}$, $C_{p}$, and $L_{b}$. The DIE curve in Figure 5 is the input-referred noise of the TIA without the capacitance $C_{d}$, $C_{p}$, and $L_{b}$, i.e., the input-referred noise of the fabricated die. EDIE is the encapsulated TIA, and the curves represent the simulated and measured input-referred noise of the TIA with packaging parasites, i.e., $C_{p}$ and $L_{b}$. Finally, CDIE is the encapsulated TIA die with a capacitance of 5 pF, $C_{d}$ at the input port. The simulation results shown in Figure 5a,b were obtained for 3000 Monte Carlo runs and three-sigma parameter variations in the given CMOS process. The simulation results were post-processed and the average values calculated and only a 3% variation in the input-referred noise was found, which is clearly an underestimation in comparison with measurements.

In order to minimize the parallel noise contribution, transistor MN4 operates at a relatively low bias current. The reduced input impedance is achieved by the booster amplifier—stage MN2-MP2 in Figure 2. In addition, the booster amplifier guarantees the control of the gain. The noise optimization is carried out through the proper balancing of the series and parallel noise source contributions from transistors MN2 and MN4. In summary, for a regulated cascode, the input-referred noise is minimized, and the input impedance is reduced. However, there is some tradeoff between noise and impedance optimization. The main drawback of operating transistor MN4 at a low bias is that high input signals modulate the transconductance $g_{mN4}$. This transconductance is part of the input impedance of the preamplifier. A change in the input impedance results in signal reflections.

### 2.2. Buffer Design for Low-Power Consumption and Signal Distortion

Signal integrity is a major problem in system-on-chip (SoC) designs, and the voltage conveyor technique has been investigated and proposed to operate at hundreds of megahertz and at low voltages. Once $v_{o}\left(s\right)/i_{in}\left(s\right)$ and the input-referred noise of the preamplifier have been discussed in detail, in this section the design of a buffer is considered. The transimpedance gain of the TIA is $v_{ot}\left(s\right)/i_{in}\left(s\right)$, which is obtained by
(8)$$\frac{v_{ot}\left(s\right)}{i_{in}\left(s\right)}=\frac{v_{ot}\left(s\right)}{v_{o}\left(s\right)}\;\frac{v_{o}\left(s\right)}{i_{in}\left(s\right)}.$$

Note that all noise is considered to be white and non-correlated to simplify the analysis. Due to the presence of the parallel equivalent noise source, in the following stages we had to filter out low-frequency noise components, and in order to improve the SNR, the upper frequency bandwidth had to be limited. Between those filter stages and the preamplifier stage, a voltage buffer was introduced; therefore, a VCII structure (see Figure 1) was obtained.

Figure 3 shows a voltage buffer implementing voltage mirroring between nodes X and Z. Transistor MN9 acts as a flipped voltage follower (FVF) [16] to obtain a fixed drain-source voltage. The voltage buffer proposed in this paper is illustrated in Figure 6. This buffer uses three diode-connected transistors (MN6, MP5, and MP9) as current sources and two non-inverting buffers (MN7 with MP7, and MN8 with MP8) and a common-drain amplifier biased by a current mirror structure (MN9–MN11). Transistor MP6 acts as an active load for the biasing of the first non-inverting buffer (MN7 and MP7).

The output currents of both non-inverting buffers are added to node 7 to drive the gate of transistor MN10. In addition, the drain current of transistor MN4 is chosen as reference current to be applied to the output current mirror (MN9–MN11). In this case, the value of resistors R1 and R2 is 500 kΩ; and that of Rx and Rz are 10 kΩ and 10 MΩ, respectively. Thus, node X conveys similar a low voltage swing to that of node Z. The buffer stage is optimized for low distortion and low energy or power consumption. As shown in Figure 6, the common drain amplifier (MP9, MN10) is driven by two non-inverting buffers for a low-distortion design.

Table 1 provides the transistor sizes for two approaches, named $DCVC_{LP}$ and $DCVC_{LD}$, the low-power (LP) and low-distorsion (LD) versions of the down-converter voltage conveyor (DCVC), respectively. The preamplifier stage was sized according to low-power and low-noise requirements, and in both approaches we considered the same stage; i.e., MP1, MP2, MP3, MN1, MN2, and MN3 had the same transistor sizes in $DCVC_{LP}$ and $DCVC_{LD}$. The rest of the transistors in Table 1 constitute the voltage buffer. When the DCVC voltage buffer was optimized for low-power consumption and low distortion, the transistor sizes were as shown in column 3 and column 6, under the set $DCVC_{LP}$ and $DCVC_{LD}$, respectively. Figure 7 illustrates the distortion in the time domain for $DCVC_{LP}$ and $DCVC_{LD}$. The active area for $DCVC_{LP}$ was 37.6% lower than that for $DCVC_{LD}$.

For distortion analysis, instead of a current stimulus, a 0.2 Vp input signal on node Y was considered. The voltages of the output nodes X and Z are shown in Figure 7. As indicated by the simulated outputs of $DCVC_{LP}$ in Figure 7, the distortion level increased with the input signal level. In contrast, $DCVC_{LD}$ presented a more lineal output and it did not suffer distortion. The total harmonic distortions were calculated through simulation and they are illustrated in Figure 8. As shown, for the first ten harmonics of $DCVC_{LD}$, the THD results were roughly five times lower than those of the $DCVC_{LP}$ solution.

The energy-delay product, as a figure-of-merit (FoM), was estimated in both circuits with an input signal of 1.1 GHz. Table 2 provides the FoMs for $DCVC_{LP}$, $DCVC_{LD}$, and $DCVC_{LDm}$—the low-power consumption version of DCVD, the low distortion version of DCVC, and the fabricated DCVC, respectively. As shown, $DCVC_{LD}$ showed increased power consumption with improved the signal integrity. The power consumption and delays of $DCVC_{LDm}$ were average values, in terms of the measurements of the non-encapsulated die of the transimpedance amplifier. For power consumption estimations, the worst case corner was used in the simulations.In this case, Table 2 illustrates that the EDP for $DCVC_{LP}$ was reduced by 44% compared to $DCVC_{LD}$. However, as shown in Table 2, the measured power consumption of the implemented version of $DCVC_{LD}$, referred to as $DCVC_{LDm}$, increased by 2.2%.

## 3. Chip Implementation

The layout used for the DCVC is shown in Figure 9. Its core area was 10.85 × 13.10 μm^2^ (H × W), that is, 142.13 μm^2^. Additionally, in Table 1, column 6, under the column designated as $DCVC_{LD}$, details of the transistor sizes can be found. The total active area in this case was 23.19 μm^2^.

As mentioned previously, our proposed TIA based on VCII was fabricated using the 65 nm low-leakage and low-K CMOS technological process for mixed-signal systems. The transimpedance amplifier was placed on the front end of a particle detector including a digital signal processor. Resistors R1, R2, and Rx were connected at input pad X. Resistor Rz was connected at output pad Z. The layout of the transimpedance amplifier was connected to a binding pad of 47 μm × 72 μm (width × length). Figure 10b shows a die photograph of the fabricated chip and the chip wire-bonded to a ceramic package. The figure illustrates the layout integration of the TIA and bonding wire connections. Two microprobes were used to measure both the low-frequency response of the package pins to the input pads and the transimpedance gain.

Figure 10 shows a die photograph of the fabricated chip close to the analog input and output PAD area. The TIA was an analog block that was on the periphery, i.e., it was on the IO of the chip at 2.4 V (±1.2 V). The rest of the chip was used for digital signal processing at 1.0 V. Voltage for powering the TIA was supplied by dedicated Vdd and Vss pads. The total area of the input and output pads and the TIA was 82 μm × 105 μm (H × W).

## 4. Measurements and Comparison Results

Figure 11a shows the measurements of the scattering parameter $S_{21}$ for one representative sample of the non-encapsulated die (DIE), i.e., it excludes the detector capacitance $C_{d}$, the bonding wire inductance $L_{b}$, and the packaging capacitance $C_{p}$. The measured $S_{21}$ ranged from 16 dB @ 1 MHz to 12.8 dB @ 1.5 GHz and 13.8 dB at the bandwidth frequency of 1.1 GHz. The transimpedance measurements for the DIE are illustrated in Figure 11b. These were obtained for post-processing measurements of 12 samples of the DIE. As shown in Figure 11b, the transimpedance frequency response was tested from 1 MHz to 1.2 GHz using the network analyzer. The low-frequency transimpedance gain was greater than 50 dB and the −3 dB bandwidth was about 1.1 GHz. The simulation results and the experimental measurements of the transimpedance gain matched very well. Group delay was calculated based on phase measurements and simulations and, as shown, it remained constant up to approximately 100 MHz. Transimpedance gain measurements and group delay simulations were obtained for more than 2000 Monte Carlo runs, with parameter variations of two-sigma; variations of ±1% in voltage supply, as specified in the technology process for 2.5 V IO transistors; and temperature variations ranging from nominal to 125 °C.

### Comparison with State of the Art

Finally, Table 3 compares the simulated (pre-layout) and measured performance of other implementations with that of our design, DCVC. For an input signal of 1.1 GHz, the measured power consumption was approximately 55.3 mW. DCVC was also driven with a square-wave digital signal, with a level between ±1 V, at 500 MHz, and the measured power consumption was close to 25 mW. Both bandwidth (BW) in GHz and transimpedance gain (G) in dBΩ were obtained from Figure 11b, using the average values at each frequency. The input-referred noise was obtained from Figure 5 using the average values. Area (A) is another performance criterionwhen the analog block is close to input/output pads, or even when it is integrated into an analog input-output pad. As shown in Table 3, depending of the performance, some approaches introduce advantages, and between some performance criteria there are tradeoffs. A common figure of merit (FoM) that combines performance in terms of G, BW, P and A is shown in the comparison. This FoM combines the tradeoffs (G × BW) and (P × A), and it represents the product G × BW, obtained per unit of area and power consumption. Therefore, higher G × BW and lower P × A correspond to a greater FoM. Another FoM for considering the tradeoff between performance and input-referred noise is FoM/N. The greater the FoM, and the lower the input-referred noise N, the higher the ratio of FoM/N, obviously.

## 5. Conclusions

In conclusion, a transimpedance amplifier based on a second-generation voltage conveyor topology, used as an interface for charge detection applications and a high-performance signal processing systems in ASIC, was successfully implemented and tested. A current buffer (CB) based on a regulated cascode topology was optimized for input-referred noise and product gain-bandwidth. Input-referred noise was analyzed with a detector capacitance $C_{d}$ placed at the input of the CB and considering both the bonding wire $L_{b}$ and packaging capacitances. The simulation and measurement results were in agreement, and they showed a noise capacitive peak which was diminished by the input inductance $L_{b}$. The voltage buffer (VB) was designed for low distortion and power consumption, as well as functioning as an impedance adaptor. The CB-VB set, i.e., the transimpedance amplifier (TIA) acting as a VCII, as well as its low distortion characteristics, demonstrate its usefulness as a linear voltage amplifier as well. The TIA was implemented in CMOS using a 65 nm logic and a mixed-mode technology process, together with a signal processor for charge detection applications, and its reduced area of 105 × 82 μm × μm represents an advantage in terms of its integration with an input–output analog pad. Based on the measurement results of 12 die samples, a 52 dB transimpedance gain and 1.1 GHz of bandwidth, together with 22 pA/√Hz of input-referred noise and 55.3 mW of power consumption, demonstrated the success of the system integration. Comparisons with published results demonstrated an improvement in terms of the figure of merit of transimpedance gain × bandwidth per unit of area, power consumption, and noise.

## Figures and Tables

**Figure 1 sensors-22-05997-f001:**
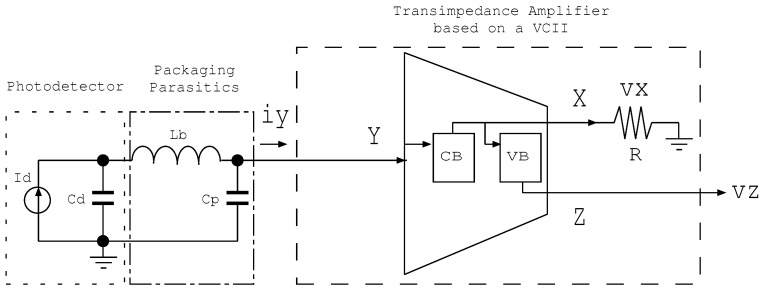
Schematic of a TIA based on VCII, including the packaging of parasitic elements as front-end receivers for converting currents to voltages.

**Figure 2 sensors-22-05997-f002:**
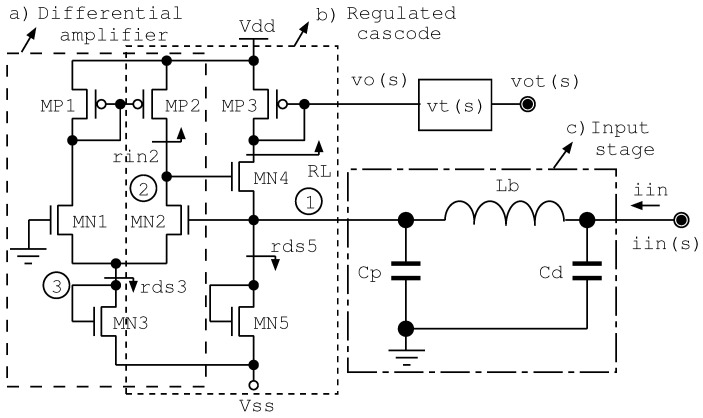
Schematic at the transistor level of the preamplifier stage. (**a**) differential amplifier; (**b**) regulated cascode; (**c**) input stage, including $C_{d}$ (diode capacitance), $L_{b}$ (wire bond inductance), and $C_{p}$ (parasitic capacitance of the pad). By the Kirchhoff’s voltage equations in nodes 0, 1, 2 and 3, the transimpedance transfer function is obtained.

**Figure 3 sensors-22-05997-f003:**
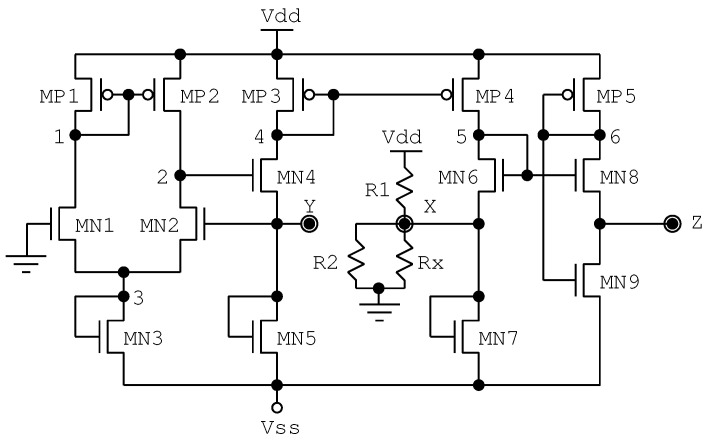
Reference circuit diagram based on [13], including modification of the current sources (SCVC).

**Figure 4 sensors-22-05997-f004:**
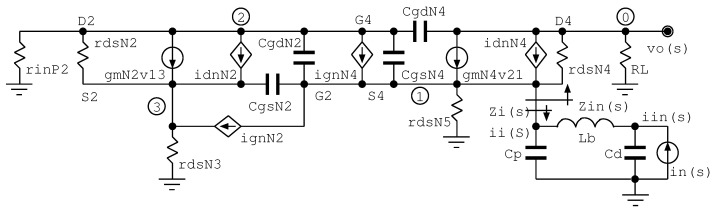
Simplified small-signal model of the preamplifier stage, including noise current sources of transistors MN2 and MN4. By the Kirchhoff’s voltage equations in nodes 0, 1, 2 and 3, the transimpedance transfer function is obtained (see Appendix A).

**Figure 5 sensors-22-05997-f005:**
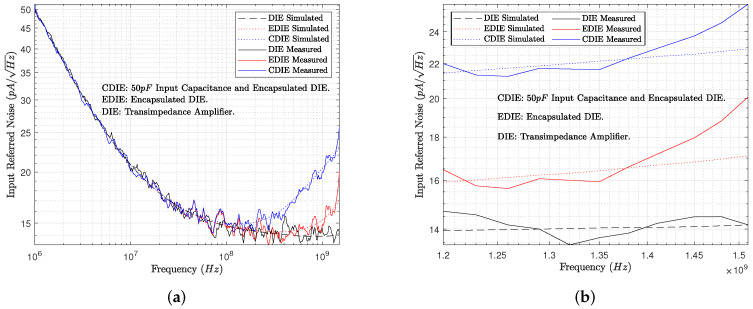
Simulated and measured input-referred noise of the TIA. (**a**) From 1 MHz to 1.2 GHz. (**b**) From 1.2 GHz to 1.5 GHz.

**Figure 6 sensors-22-05997-f006:**
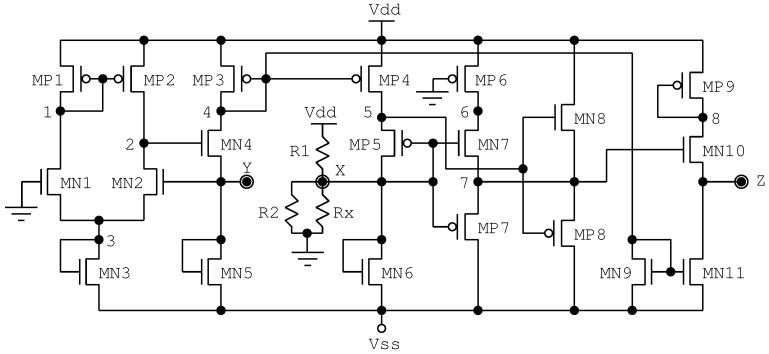
Schematic for the proposed TIA.

**Figure 7 sensors-22-05997-f007:**
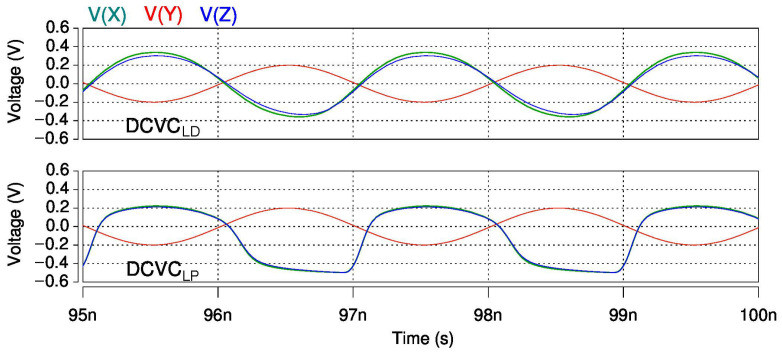
Inputs and outputs waveforms for $DCVC_{LD}$ and $DCVC_{LP}$.

**Figure 8 sensors-22-05997-f008:**
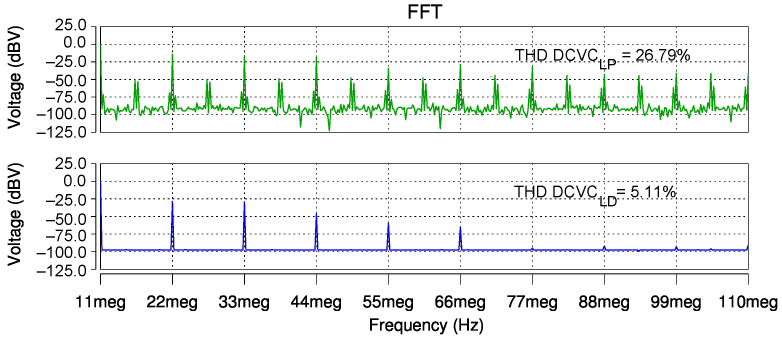
Post-layout THD comparison for $DCVC_{LP}$ and $DCVC_{LD}$.

**Figure 9 sensors-22-05997-f009:**
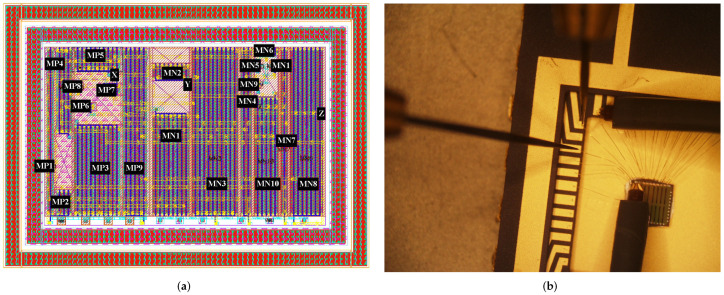
Layout used for the DCVC circuit in 65 nm CMOS technology; (**a**) designed layout and (**b**) encapsulated die.

**Figure 10 sensors-22-05997-f010:**
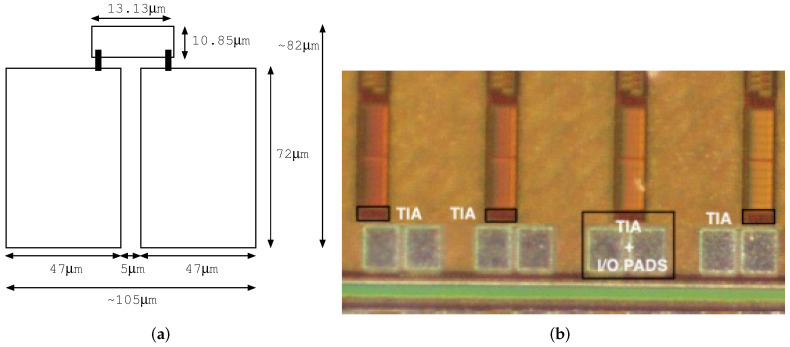
Die photograph of the input and output analog pad area, including the layout of the transimpedance amplifier. (**a**) Relevant dimensions of TIA and I/O pads, and (**b**) microphoto of the fabricated die.

**Figure 11 sensors-22-05997-f011:**
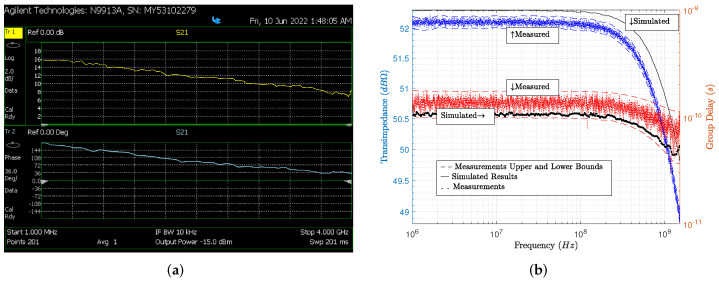
Measured frequency response of DIE: (**a**) S21 s-parameter, where magnitude and phase are colored in yellow and blue, respectively; and (**b**) transimpedance and group delay in blue and red, respectively.

**Table 1 sensors-22-05997-t001:** Sizing of PMOS and NMOS transistors.

${\mathrm{DCVC}}_{\mathrm{LP}}$			${\mathrm{DCVC}}_{\mathrm{LD}}$		
Label	Type	Width (μm) ${}^{*1}$	Label	Type	Width (μm) ^∗2^
MP1	P	0.7 × 10	MP1	P	0.7 × 10
MP2	P	0.45 × 10	MP2	P	0.45 × 10
MP3	P	5.3 × 10	MP3	P	5.3 × 10
MN1	N	5.0 × 10	MN1	N	5.0 × 10
MN2	N	1.0 × 10	MN2	N	1.0 × 10
MN3	N	1.5 × 10	MN3	N	1.5 × 10
MN4	N	2.0 × 10	MN4	N	2.0 × 10
MN5	N	0.025 × 10	MN5	N	0.025 × 10
MP4	P	1.5 × 2	MP4	P	7.5 × 2
MP5	P	1.0 × 10	MP5	P	1.2 × 10
MP6	N	2.0 × 0.25	MP6	P	4.0 × 0.25
MP7	N	0.2 × 4	MP7	P	0.5 × 4
MP8	N	0.1 × 1	MP8	P	0.25 × 1
MP9	N	2.5 × 10	MP9	P	5.0 × 10
MN6	N	0.13 × 5	MN6	N	0.26 × 5
MN7	N	0.5 × 10	MN7	N	1.0 × 10
MN8	N	3.0 × 10	MN8	N	7.0 × 10
MN9	N	0.25 × 1	MN9	N	0.25 × 1
MN10	N	3.5 × 7	MN10	N	5.0 × 7
MN11	N	0.35 × 1	MN11	N	0.35 × 1

*^1^: Active area = 16.90 μm^2^. *^2^: Active area = 23.19 μm^2^. Vthn = 0.28 V, and Vthp = −0.2 V.

**Table 2 sensors-22-05997-t002:** Power, energy, delay, and energy-delay-product (EDP) obtained using a signal input of 0.2 Vp @ 1.1 GHz.

VC	Power	Energy	Delay	EDP
Approach	mW	pJ	ps	pJ × ns
$DCVC_{LP}$	25.27	23	92	2.12
$DCVC_{LD}$	54.12	49	77	3.76
$DCVC_{LDm}$	55.3	50	79	3.95

**Table 3 sensors-22-05997-t003:** Comparison between circuit design approaches proposed in the literature.

Design	[20]	[18]	[14]	[13]	Ours
Year	2014	2016	2019	2020	2022
Name	RCG	TSCG	VCII	VCII	DCVC
Technology	CMOS 130 nm	CMOS 180 nm	CMOS 180 nm	Discrete	CMOS 65 nm
Results *^1^	Meas	Meas	Pre	Meas	Meas
Supply V	1.2	1.8	±0.9	±5	±1.2
Area (A) μm × μm	540 × 410	1133 × 1283	NA	NA	105 × 82
Power (P) mW	0.34	48.6	0.179	200	55.3
Bandwidth (BW) GHz	0.01	1.75	0.169	0.106	1.1
Gain (G) dBΩ	100	83	60	42	52
Noise (N) pA/Hz^1/2^	2.7	2.4	NA	55	22
FoM $\left(\frac{{\left(G\times BW\right)}^{1/2}}{P\times A}\right)$	13.27 × $10^{-9}$	171.1 × $10^{-9}$	NA	NA	6.95 × $10^{-6}$
FoM/N	NA	71.29 $\times 10^{-9}$	NA	NA	315.9 × $10^{-9}$

NA: Not available. *^1^ Pre: Pre-layout. Meas: Measured.

## Data Availability

Not applicable.

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
