# Peer review of "High Gain, Low Noise and Power Transimpedance Amplifier Based on Second Generation Voltage Conveyor in 65 nm CMOS Technology"

_sensors, 2022, doi:10.3390/s22165997_

Round 1

Reviewer 1 Report

In this paper, authors present a transimpedance amplifier (TIA) based on a voltage conveyor structure designed for high gain, low noise, low distortion, and low power consumption. Following a second generation voltage conveyor topology, the current and voltage blocks are a regulated cascode amplifier and a down converter buffer, respectively. The proposed voltage buffer is designed for low distortion and low power consumption, whilst the regulated cascode for low noise a high gain. The resulting TIA has been fabricated in 65 nm CMOS technology for logic and mixed-mode design, using low threshold voltage transistors and a supply voltage of ±1.2 V. It exhibits a 52 dBΩ transimpedance gain and a 1.1 GHz bandwidth while consuming 127 mW using ±1.2 V supply.

Generally speaking, althought the proposed scheme is based on circuit introduced in [37], therefore the novelty is quite limited, the paper is well-written, structured and it is complete of design procedure and extensive measurement results.

I have no particular comments or suggestions of the authors and their work that, in my opinion, is well-done.

Reviewer 2 Report

The paper presents a fast TZA amplifier in a 65nm technology, with experimental results.

After a long introductory part on a literature topology, the proposed topology is shown in Fig. 6.

This topology has a huge number of problems which need to be addressed.

1. Mn3,5,6 are diodes and provide very small resistance, which is not even copied from node Y to node X because the difference in currents between MN5 and MN6 depend on the respective voltages. Hence, the current buffering is poor.

2. MN7-MP6 have a common drain node (6), whose voltage is thus arbitrary, and may create biasing issues.

3. The entire TIA exploits 2.4V supply voltages, while the breakdown voltage is 1.2V, so that many nodes (especially those with only two or three piled devices) may cause breakdown in the relative devices.

4. MN8-MP8 form a class-B output stage with no cross-over distortion compensation, whose biasing current will be 0 most of the time.

5. MP3 and MN9 form two series diodes, each with about 1.2V Vgs.

Given the issues with the topology, I have doubts the VCII can work reliably under PVT and Monte Carlo (process + mismatch) variations. No simulations are reported, so that it is not possible to judge whether the circuit is robust. The VCII should be fully characterized in terms of biasing point, AC simulations and signal swings to convince the reader that experimental results are robust and not just due to chance or extensive tuning of the setup.

The first row in Fig. 8 shows many distortion spikes: a sinusoidal input signal should provide a spectrum similar to the second row. How was the first row obtained?

It is strange that most of the theoretical analysis is about the standard circuits in Figs. 2 and 3, and little is said about the proposed circuit in Fig. 6.

All the papers in Table 2 use nominal supply voltages, whereas the proposed topology uses twice the maximum supply voltage for the employed process.

What is 'a' in Eq. 1?

Reviewer 3 Report

The Authors present a transimpedance amplifier topology based on a VCII, to design an integrated front-end for particle detector. The work and the results are valuable and interesting for the reader, even if the VCII is not completely new: the voltage buffer can be found in the literature, as pointed out by the Authors themselves.

Some minor modifications are needed:

- please explain the meaning of Low-K process

- node equations 2-5 are not essential, they could be deleted or put in Appendix

- lines 147 (not numbered), the value of the input impedance correlation in comparison against common gate topology: please explain better

- lines 225-228, named DCVCLP and DCVCLN. The preamplifier stage has been sized for low-power and low-noise, and in both approaches are considered: please, define the two approaches DCVCLP and DCVCLN.

Moreover, English language needs improvements:

- line 6, for low noise a high gain: AND?

- line 16: demonstrate

- line 33, despite the fact that their low cost

- line 49, In particular to detectors, it is used two topologies

- line 68, en the connected

- line 73, two VC

- line 85, and also reach a working bandwidth of 106 MHz and 200 mW power consumption

- line 87, reducing of both

- line 96, Simulation and measurement results shows

- line 151 and 291, As it is shown

- line 195, is achieved by the of the

- line 197, controlled of the gain.

- line 202, modulate

- lines 204+2 (not numbered), operate at several 100 MHz speeds (?)

- line 206, following stages has to filter

- line 215, by a current the mirror structure

- line 229, The rest of the transistors

- line 236, The outputs drives nodes X and Z as are shown

- line 238, presents a correct lineal output

- line 240, and is illustrated

- line 255, he transistor sizes is found

- line 265, two TIA outputs

- line 290, as is the case in

- line 291, As is shown in Table 2 , no space

- lines 304-305, for both noise and bandwidth gain: not clear

- line 305, a 50 dB transimpedance gain
